# NLRP3 Inflammasome Inhibitors for Antiepileptogenic Drug Discovery and Development

**DOI:** 10.3390/ijms25116078

**Published:** 2024-05-31

**Authors:** Inamul Haque, Pritam Thapa, Douglas M. Burns, Jianping Zhou, Mukut Sharma, Ram Sharma, Vikas Singh

**Affiliations:** 1Research and Development Service, Kansas City Veterans Affairs Medical Center, Kansas City, MO 64128, USA; pritam.thapa@va.gov (P.T.); dmburns9@yahoo.com (D.M.B.); mukut.shrma@va.gov (M.S.); ram.sharma2@va.gov (R.S.); 2Department of Math, Science and Business Technology, Kansas City Kansas Community College, Kansas City, KS 66112, USA; 3Department of Cancer Biology, University of Kansas Medical Center, Kansas City, KS 66160, USA; 4Drug Discovery Program, Midwest Veterans’ Biomedical Research Foundation, KCVA Medical Center, Kansas City, MO 64128, USA; 5Renal Research Laboratory, Kansas City VA Medical Center, Kansas City, MO 64128, USA; jianping.zhou@va.gov; 6Division of Neurology, Kansas City VA Medical Center, Kansas City, MO 64128, USA

**Keywords:** antiseizure medication, ASC, caspase-1, epilepsy, epileptogenesis, interleukin-1β, neuroinflammation, NLRP3 inflammasome

## Abstract

Epilepsy is one of the most prevalent and serious brain disorders and affects over 70 million people globally. Antiseizure medications (ASMs) relieve symptoms and prevent the occurrence of future seizures in epileptic patients but have a limited effect on epileptogenesis. Addressing the multifaceted nature of epileptogenesis and its association with the Nod-like receptor family pyrin domain containing 3 (NLRP3) inflammasome-mediated neuroinflammation requires a comprehensive understanding of the underlying mechanisms of these medications for the development of targeted therapeutic strategies beyond conventional antiseizure treatments. Several types of NLRP3 inhibitors have been developed and their effect has been validated both in in vitro and in vivo models of epileptogenesis. In this review, we discuss the advances in understanding the regulatory mechanisms of NLRP3 activation as well as progress made, and challenges faced in the development of NLRP3 inhibitors for the treatment of epilepsy.

## 1. Introduction

Epilepsy is a serious chronic brain disorder characterized by spontaneous and recurrent seizures [1,2] that affects over 70 million people worldwide [3]. Approximately 3 million adults and 470,000 children suffer from epilepsy, with an incidence of about 150,000 new cases each year in the USA [4]. According to the World Health Organization, roughly half of epilepsy cases have no known cause, whereas the other half may be due to imbalance of neurotransmitters, brain tumors, strokes, immune disorders, gene mutations, and traumatic brain injury. The majority of epileptic patients also suffer from depression and memory loss, resulting in a poor quality of life and reduced life expectancy [5]. The use of antiseizure medications (ASMs) remains the preferred treatment, and seizures are being effectively controlled in 70% of epileptic patients [6]. Although more than 30 FDA-approved ASMs with diverse molecular targets are available for therapeutic purposes [7], the biggest challenge in the clinical treatment of epilepsy is that about 30% of epileptic patients develop resistance to ASMs. There is an urgent need to develop newer drugs with better efficacy, fewer side effects, and less toxicity. This need is further augmented by the fact that available ASMs do not address epileptogenesis, the disease process that leads to epilepsy [8,9]. Understanding of the mechanisms involved in epileptogenesis will be crucial in the development of novel drugs that target the epileptogenic process.

## 2. Epileptogenesis

Epileptogenesis is a poorly understood, multifaceted chronic process that alters the normal brain into an epileptic brain capable of generating spontaneous recurrent seizures (SRSs) [10]. Some of the common processes involved in epileptogenesis include neuroinflammation, neuronal cell death in the temporal lobe and hippocampus, bioenergetic dysfunction, altered neuroplasticity, blood–brain barrier (BBB) disruption, astrogliosis, microglial activation, and the infiltration of inflammatory cells into brain parenchyma. Injections of a chemo-convulsant, such as pilocarpine or kainic acid, or repetitive electrical stimulation of the hippocampus or amygdala, can result in a chronic epileptic condition characterized by strong convulsive SRSs in animal models of acquired epilepsy, the use of which will help tremendously in better diagnosis, treatment, and prevention of human epilepsy. Anticonvulsant drugs mostly relieve symptoms and prevent future seizures in patients with epilepsy rather than inhibiting and/or modulating the neuroinflammatory processes of epileptogenesis [9]. The development of targeted therapeutic strategies beyond conventional anticonvulsant/antiseizure treatments is needed to address epileptogenesis and epilepsy.

In general, neuroinflammation is beneficial because it promotes homeostatic neurogenesis [11,12], provides protection against the loss of axons and neurons [13], and promotes axonal regeneration [14,15], but chronic neuroinflammation has been shown to be detrimental in numerous neurological disorders including Alzheimer’s disease, meningitis, Parkinson’s disease, multiple sclerosis, encephalitis [16], and epilepsy [17,18,19]. Despite the absence of pathogens, neuroinflammation in epilepsy is triggered by an abnormal increase in proinflammatory mediators [20] or by endogenous inducers (damage-associated molecular patterns [DAMPs]) produced and secreted by brain cells undergoing stressful events termed as sterile inflammation [21]. Damage-associated molecular patterns such as adenosine triphosphate (ATP), high mobility group box1 (HMGB1), reactive oxygen species (ROSs), K^+^ efflux, advance glycation end products (AGEs), S1100b (a Ca^2+^ binding protein), or migration inhibitory factor-related protein 8 (MRP8) are known to induce the formation of inflammasome, a multi-molecular protein complex [22,23,24].

## 3. Inflammasomes

In 2002, Tschopp and colleagues coined the term “inflammasome” to describe a high-molecular-weight protein complex that mediates the activation of inflammatory caspases [25]. An inflammasome is comprised of a sensor protein (a cytoplasmic pattern-recognition receptor (PRR) classified by its protein domain structures), an adapter protein (apoptosis-associated speck-like protein containing a CARD [ASC]), and pro-caspase-1 as an effector protein [26]. Typically, inflammasomes are named after their cytoplasmic pattern-recognition receptors (PRRs), sensor proteins that are either of a NLR family (a nucleotide binding domain (NBD) combined with a leucine-rich-repeat-containing receptor (LRR-CR) protein) or an ALR family (an absence in melanoma 2 (AIM2)-like receptor (ALR) protein). Most pertinent inflammasomes belong to the NLR family subgroups, containing pyrene domain (NLRP) or CARD domain (NLRC) proteins. The best known inflammasomes are NLRP1; NLRP2; NLRP3; NLRP6, NLRP7; NLRP12; NLR-family apoptosis inhibitory protein (NAIP); NLR-family, CARD-containing 4 (NLRC4); AIM2; and Pyrin inflammasomes. Numerous studies have demonstrated that the central nervous system (CNS) is equipped with different inflammasome subtypes; however, their expression profiles vary among CNS cell types [27,28,29,30,31]. NLRP1 is mostly expressed in neurons and oligodendrocytes [32,33,34,35], NLRP2 in astrocytes [36,37], NLRP3 in microglia [38,39,40,41], NLRC4 in astrocytes [42,43], and AIM2 in neurons [44,45]. NLRP3, also known as cryopyrin, is the most widely studied and best characterized inflammasome. The NLRP3 protein has an N-terminal Pyrin domain (PYD), a central ATPase domain known as NACHT that comprises the NBD, a helical domain 1 (HD1), a winged helix domain (WHD), a helical domain 2 (HD2), and a C-terminal LRR domain [46].

NIMA (never in mitosis A)-related kinase 7 (NEK7) is a serine/threonine kinase that modulates microtubule stability, mitotic spindle formation, and cytokinesis and has now been recognized as a crucial component of the NLRP3 inflammasome [47,48]. NEK7 senses reactive oxygen species (ROSs), K^+^ efflux, lysosomal destabilization, and nuclear factor kappa-light-chain-enhancer of activated B cells (NF-κB) signaling-mediated activation of NLRP3 [49,50]. A cryo-EM structure of the NLRP3-NEK7 protein complex suggests that the first half of the NEK7 C-terminal interacts with the LRR domain of NLRP3 and the second half with the NBD and HD 2 domains of NACHT [46]. The binding of NEK7 alone is not sufficient to activate NLRP3. In addition, ATP needs to bind to the NBD domain and then be used to phosphorylate S195 in NEK7, prompting NLRP3 to adopt a modified conformation that drives oligomerization and activation of the NLRP3 complex [51]. Thus, NEK7 plays a licensing role in NLRP3 activation rather than serving as the sole activator [46].

## 4. Activation of the NLRP3 Inflammasome

The physiological activation of NLRP3 is tightly regulated to avoid uncontrolled inflammatory responses and is beneficial against invading pathogens and tissue damage, whereas abnormal or chronic activation plays a detrimental role in cellular and body health. As depicted in Figure 1, NLRP3 inflammasome activation can occur via canonical, non-canonical, and alternative pathways [52]. Optimal activation of NLRP3 via canonical pathway requires priming step/Signal 1, initiated by the detection of extracellular PAMPs and endogenous cytokines by the PRRs, which then upregulates the NF-κB-mediated transcription of NLRP3, pro-IL-1β, and pro-IL-18 [53]. In addition, priming also regulates NLRP3 at the post-transcriptional level through DNA methylation, protein acetylation, and the use of microRNAs. For example, Wei et al. reported that NLRP3 expression was increased through NLRP3 promoter demethylation in response to *Mycobacterium tuberculosis* infection [54]. It has been revealed that the NLRP3 induction by lipopolysaccharide (LPS) priming is decreased by binding of the RNA-binding protein Tristetraprolin [55] and miRNA-223 [56] to 3′UTR of NLRP3. Moreover, priming also involves various post-translational modifications (PTMs) of the NLRP3 protein, such as ubiquitination and phosphorylation, that silence NLRP3 for subsequent activation by a second signal.

The second activation step, or Signal 2, is the assembly of the inflammasome proteins triggered by sensing intracellular PAMPs/DAMPs through a combination of important and related events such as activation of the purinergic receptor P2X7 by ATP, cathepsin release following lysozyme rupture, the opening of Ca^2+^ channels to allow ion flux, mitochondrial dysfunction, ROS formation, Golgi apparatus disassembly, and endoplasmic reticulum stress (ERS) [57]. Once activated, oligomerization of the NLRP3 inflammasome involves two homotypic protein–protein interactions. First, the PYD domain of NLRP3 protein interacts with the adaptor protein ASC via PYD–PYD interactions and the CARD domain of ASC interacts with the CARD domain of pro-caspase 1 to recruit it to the NLRP3 inflammasome [58]. The activated NLRP3 inflammasome is thought to induce conformational changes that generate active caspase-1 (Casp1), which converts pro-interleukin-1β (pro-IL-1β) and pro-interleukin-18 (pro-IL-18) to mature bioactive IL-1β and IL-18 [53]. Additionally, Casp1 cleaves the protein gasdermin D (GSDMD) to generate N-terminal GSDMD (N-GSDMD) [59]. After cleavage, N-GSDMD oligomerizes in the cell membrane to form pores, allowing IL-1β and IL-18 to leave the cell and effectively execute a highly inflammatory form of cell death that is termed pyroptosis [25,60].

In the non-canonical pathway for NLRP3 inflammasome activation, intracellular LPS that has been generated by Gram-negative bacteria is directly recognized by the CARD domain of caspase-11 in mice and caspase-4/5 in humans, ultimately leading to IL-1β and IL-18 release through the activation of the NLRP3-ASC-Casp1 pathway [61,62,63,64].

The third or the alternate pathway of NLRP3 inflammasome activation exists in both human and porcine monocytes, but not in murine cells [65]. In this alternate pathway, NLRP3 assembly occurs upon the activation of toll-like receptor 4 (TLR4) in response to LPS and requires the TIR-domain-containing adapter-inducing interferon-β (TRIF)-receptor-interacting serine/threonine-protein kinase 1 (RIPK1)-Fas-associated protein with death domain (FADD)-caspase-8 signaling cascade to activate the NLRP3 inflammasome, leading again to the cleavage of pro-IL-1β to produce mature and active IL-1β [66]. In contrast to classical pathways, the alternative pathway lacks ASC speck formation, pyroptosis induction, or K^+^ efflux [67].

## 5. NLRP3 Inflammasome Involvement in Epilepsy

An increase in NLRP3 activity has been shown in the brains of individuals with epilepsy and in animal models of epilepsy [68,69,70,71,72,73,74,75]. Meng et al. first documented a significant increase in NLRP3 inflammasome in the amygdala kindling-induced murine model of seizures [70]. NLRP3 inflammasome activation in kainic acid (KA)-induced epileptic rats became evident when significant neuronal loss was noticed along with elevated levels of mature IL-1β, active caspase 1, and expression of the NLRP3 protein [68]. Magalhães et al. documented enhanced expression and activation of the NLRP3 inflammasome in organotypic slices as an ex vivo model of epilepsy [76]. In another in vivo study, NLRP3 inflammasome activation was significantly correlated with epileptic neuronal apoptosis [72]. Jiang and colleagues revealed that an increase in NLRP3 gene transcription is due to binding of Stat3 acetylation of the histone H3K9 site of the NLRP3 promoter resulting in increased NLRP3/caspase-1-mediated hippocampal neuronal cell death in epileptic mice [77]. NLRP3 inflammasome activation has also been reported in the hippocampus of pentylenetetrazole (PTZ)-induced epileptic mice and LPS-induced BV2 microglial cells [78]. Yue and colleagues revealed that the NLRP3 inflammasome components are highly expressed in neurons, microglia, and astrocytes in the epileptogenic zone of patients with temporal lobe epilepsy (TLE), a form of epilepsy associated with hippocampal neuronal atrophy, and in the hippocampi of a mouse model of status epilepticus (SE)-induced by pilocarpine [73]. Furthermore, they also demonstrated a positive correlation between increased levels of NLRP3 inflammasome proteins (NLRP3, ASC, and Caspase1) and ERS-related protein markers (GRP78, PERK, p-PERK, eIF2a, p-eIF2a, ATF4, and CHOP) in patients with TLE. Additional evidence suggests that persistent ERS may induce inflammatory processes, leading to seizures [79]. In another study, they showed that levels of RevErba, a putative down regulator of the NLRP3 inflammasome, is diminished in the leptogenic area of patients with mesial TLE (mTLE) [80]. Wu and colleagues reported that the number of NLRP3^+^ cells in the temporal lobe cortical tissues of refractory epilepsy patients was significantly higher than the control group [81]. They also demonstrated increased levels of NRLP3 and IL-1β in the blood of patients with refractory TLE. Furthermore, NLRP3 has been reported to be upregulated and expressed by both neuronal and glial cells in the sclerotic hippocampi of mTLE patients, which may contribute to the overexpression of hippocampal caspase-1 and IL-1β [82]. A study form Wu et al. provided the first evidence that autophagy plays a crucial role in NLRP3 inflammasome activation in the development of epilepsy [83]. NLRP3 expression was also found to be increased in children with febrile seizures [84]. A recent report by Zhang et al. confirmed that activation of NLRP3 inflammasome enhances the expression of adenosine kinase to accelerate epilepsy in mice through the CREB/REST/Spa signaling pathway [85]. Pohlentz et al. demonstrated that NLRP3 and its associated signaling molecules are activated in brain tissue samples of patients with TLE and in pilocarpine- and KA-induced SE mouse models [86].

Tan et al. demonstrated, for the first time, the elevated expression of NLRP1 and active caspase-1 in surgically resected hippocampi of patients with mTLE compared to a healthy group [33]. Furthermore, NLRP1-mediated caspase-1-dependent neuronal pyroptosis within the hippocampus was impeded in NLRP1/capase-1-deficient rats. A decreased frequency and severity of seizures was also observed in these rats. Differential gene expression analysis of the RNAseq data demonstrated the upregulation of NLRP1 mRNA in hippocampal tissues resected from patients with mTLE-hippocampal sclerosis [87]. Increased expression of NLRP1 inflammasome was observed in the sclerotic hippocampi of mTLE patients [82]. Gao et al. reported an elevated expression of the NLRP1 inflammasome in hippocampi from the pentylenetetrazol (PTZ) kindling model of epilepsy in rats [88]. Recently, the expression of NLRP1 and NLRP3 mRNAs was significantly increased in the animal model of epilepsy induced by the intrahippocampal injection of KA compared to controls [89].

## 6. Current Antiseizure Medications for the Treatment of Epilepsy

Since epilepsy is a complex multifaceted disease, no single ASM has emerged over time as the primary treatment. Instead, individualized treatment is given to a patient with a choice of ASM dependent upon the specific epileptic syndrome [6,87]. The majority of the 30 currently existing ASMs are effective and well-tolerated by patients [6], and are known to target either the GABAergic system or voltage-gated channels to curb the abnormal neuronal activity in the brain during seizures [90,91,92]. Some of these drugs are illustrated in Figure 2, emphasizing the diversity in their chemical structures. It is important to note that, even with the use of new ASMs, as many as 30% of epileptic patients still fail to benefit from ASM treatments (5,6, 132–135).

### 6.1. Currently Preferred ASM Used to Treat Epilepsy

Valproic acid, an 8-carbon 2-chain fatty acid, is the drug of choice for both adults and children suffering from epilepsy [93]. At a therapeutic range (50–100 mg/mL), it attenuates the high frequency firing of cortical and spinal cord neurons via the blockade of Na^+^, K^+^, and Ca^2+^ channels [94]. Because of valproate-induced hepatotoxicity, ethosuximide (3-ethyl-3-methyl pyrrolidine-2,5-dione) is considered a safer alternative [95]. This drug acts to disrupt the abnormal electrical activity of the thalamocortical circuitry by blocking T-type Ca^2+^ channels [91]. Clonazepam (1,3-Dihydro-2H-1,4-benzodiazepin-2-one), a long-acting benzodiazepine, can reduce the number of epileptic seizures experienced by epilepsy patients [96] by increasing neurotransmitter γ-aminobutyric acid (GABA) to decrease any abnormal electrical nerve activity in the CNS that might be contributing to seizures [97]. Clonazepam may lose its effectiveness over time due to the development of tolerance. One of the original ASMs, phenobarbital, is still used to prevent all types of seizures, but is not used to treat absence seizures. It exhibits its antiepileptic effect by increasing the inhibitory drive of GABA [98], and despite its sedative side effects, it is still widely used due to its low cost, particularly in developing countries [99]. A recent report suggests that levetiracetam (LEV), a novel ASM, exerts a neuroprotective effect by inhibiting the expression of proinflammatory molecules, such as IL-6, TNF-α, and IL-1β [100,101]. Lamotrigine, a synthetic phenyl triazine, is used alone or with other medications to treat epileptic seizures in children and adults [102]. Although its mechanism of action is not entirely understood, it appears to inhibit the release of excitatory neurotransmitters such as glutamate and aspartate triggered by voltage-sensitive Na^+^ channels and voltage-gated Ca^2+^ channels in neurons [103]. Results from Marson et al. support the continued use of lamotrigine as the best first-line treatment option for patients newly diagnosed with focal epilepsy [104]. Topiramate, a sulfamate-substituted monosaccharide, reduces the frequency and duration of seizures and is used in the treatment of certain types of epilepsy. Topiramate’s antiepileptic effect is mediated through several mechanisms, including (i) the blockade of voltage-sensitive Na^+^ channels and/or Ca^2+^-channels, (ii) the enhancement of GABA-mediated Cl^−^ fluxes into neurons, (iii) increases in K^+^ conductance, and (iv) the inhibition of glutamate-mediated neurotransmission [105]. Gabapentin, designed as a lipophilic analogue of GABA, can easily cross the BBB to increase brain synaptic GABA [106] and to suppress the influx of Ca^2+^ ions into neurons via voltage-dependent Ca^2+^ channels. Phenytoin has been used for several decades in the treatment of children with partial and generalized tonic-clonic seizures via a membrane potential-dependent blockade of Na^+^ channels. In addition to working through some of the mechanisms described above, zonisamide also reduces the concentration of free radicals, which may be effective against certain primarily generalized seizures, such as absence, tonic-clonic, and tonic seizures. [107]. Oxcarbazepine, a keto analog of carbamazepine, is a safer and more efficacious drug for treating partial onset seizures in both adult and children epileptic patients, and its antiepileptic activity is mediated by the blocking of neuronal ion channels [108]. Eslicarbazepine acetate, a third-generation ASM, has been proven effective when used in combination with other drugs to reduce the number of seizures in drug-resistant focal epilepsy [109].

There are several other drugs available for the treatment of epilepsy, which include lacosamide, brivaracetam, and perampanel [110]. Additionally, new treatments have been developed such as vagal nerve stimulation and ketogenic diets [111,112,113]. These new therapies and drugs have the potential to revolutionize epilepsy treatment, but much research is needed before they can be approved for wider use.

### 6.2. The Limitations of Current Clinical ASMs and Potential New Avenue for Drug Development in Epilepsy

Despite their effectiveness, ASMs have serious and life-threatening side effects, and a high percentage of epileptic patients develop resistance over time. Some of the generalized side effects of ASMs include depression, suicidal thoughts, mood changes, or hostility, nausea, vomiting, coordination problems, sleepiness, dizziness, and hepatotoxicity. Certain epilepsy medications can also interact with other drugs, such as pain killers or antibiotics, leading to serious consequences. The pharmacodynamic reaction of lamotrigine with carbamazepine can lead to carbamazepine intoxication [114]. Other ASMs such as phenobarbital, carbamazepine, primidone, and phenytoin can cause an imbalance in the patient’s calcium and vitamin D levels, leading to osteoporosis [115]. Due to the restrictive permeability and active efflux of some ASMs, including phenobarbital and phenytoin, the BBB limits the delivery and/or the transport of ASMs to the brain [116]. As the underlying cause(s) of epilepsy are not fully understood, the drugs developed for treating epilepsy primarily aim at controlling seizures and not at addressing the underlying cause of the disease. Given the current situation, there is a need for a more effective therapy for epilepsy.

## 7. NLRP3 Inflammasome in Epilepsy

Recent research suggest that increased activity of NLRP3 contributes to the development and progression of epilepsy [68,70,71,73,74,75,117]; hence, inhibiting the activity of NLRP3 may reduce inflammation-caused epileptic injuries and potentially improve symptoms [89]. NLRP3 inhibitors, a potential new class of drugs, might have fewer side effects and be more effective in a larger number of patients, and could target the underlying causes of epilepsy. Additionally, NLRP3 inhibitors have the potential to be effective in treating other inflammatory and autoimmune disorders such as gout, Alzheimer’s disease, and certain cancers, which makes the development of NLRP3 inhibitors a promising area of research [118,119,120,121].

There are several approaches that can be taken to develop NLRP3 inhibitors, including:

(i). Small molecule inhibitors: these are drugs that can bind to specific sites on the NLRP3 protein and prevent it from activating. These can be identified through high-throughput screening of chemical libraries; (ii). Peptide inhibitors: these are short chains of amino acids that can bind to the NLRP3 protein and inhibit NLRP3 inflammasome activation. These can be identified through phage display or other peptide-based screening methods; (iii). Antibodies: these are proteins that can bind to specific regions of the NLRP3 protein and prevent it from activating. These can be generated through antibody-based screening methods; (iv). RNA interference: this is a method for silencing specific genes by targeting their RNA. It can be used to target the NLRP3 gene and prevent it from being expressed.

Once potential NLRP3 inhibitors have been identified, they can be further tested in cell-based assays and animal models to evaluate their efficacy and safety before moving on to clinical trials.

### 7.1. NLRP3 Inhibitors in Preclinical and Clinical Trial Phase

Since the role of the NLRP3 inflammasome pathway in the pathogenesis and progression of epilepsy has been well documented, the development of NLRP3 inhibitors as a potential therapeutic target for the treatment of seizures and epilepsy is urgently needed. In epileptic animal models, the knock-down of NLRP3 has been shown to reduce neuronal cell death and attenuate the chronic seizure phenotype [70,72,81]. Inhibition of NLRP3 using the pump-mediated in vivo infusion of nonviral siRNA provides neuroprotection in rats following amygdala kindling-induced SE [70]. Recent studies have reported several inhibitors that directly or indirectly target NLRP3 inflammasome and can reduce inflammation, promote neuroprotection, and decrease seizures [122,123,124]. Some of the NLRP3 inhibitors depicted in Figure 3 could help in improving our understanding of the underlying biological mechanisms that contribute to epileptic seizures and, thus, could help in improving our ability to diagnose and treat the condition. Several clinical trials are underway, exploring the efficacy and safety of NLRP3 inhibitors in patients with chronic, drug-resistant epilepsy. Some of NLRP 3 inhibitors with their nature, mechanism of action, and disease models are shown in Table 1.

MCC950 (N-[[(1,2,3,5,6,7-hexahydro-s-indacen-4-yl) amino] carbonyl]-4-(1-hydroxy-1-methylethyl)-2-furansulfonamide), also known as CP-456, 773, or cytokine release inhibitory drugs 3 (CRID3), is best characterized as a potent NLRP3 inhibitor. It blocks both canonical and non-canonical NLRP3 inflammasome activation, but no inhibitory effect has been reported on AIM2, NLRC4, or NLRP1inflammosome activation [125]. Mechanistically, MCC950 does not inhibit the priming step of NLRP3 activation as well as K^+^ efflux, Ca^2+^ flux, NLRP3–NLRP3, NEK7–NLRP3, or NLRP3–ASC interactions [125], but it directly interacts with Walker B motif of the NACHT domain with a high-affinity non-covalent interaction, blocking NLRP3 from hydrolyzing ATP to ADP and conformational changes critical for NLRP3 oligomerization and activation [126,127]. Preclinical studies have shown promise in several different types of inflammatory diseases, including Crohn’s disease [128], ulcerative colitis [129], Alzheimer’s disease [130,131], rheumatoid arthritis [132], Huntington’s disease [133], cardiovascular disease [134], and multiple sclerosis [135]. In bone marrow-derived macrophages (BMDMs), MCC950 showed inhibition of IL-1β release at IC_50_ of 7.5 nM, while in human monocyte-derived macrophages (HMDMs), IC_50_ is 8.1 nM [125,136]. MCC950 also reduces brain injury and inflammation in a mouse model of traumatic brain injury [137]. Recent evidence showed that, in an in vitro SH SY5Y model and an in vivo model of cerebral trauma induced by PTZ, the administration of MCC950 significantly provided a protective effect, and reduced epileptic neuronal apoptosis by inhibiting NLRP3 inflammasome activation [72]. A positive correlation between NLRP3 and ERS has been observed in several models of epilepsy, including temporal lobe epilepsy, and in human brain tissues from patients with epilepsy [138,139,140,141], suggesting that it may be an underlying mechanism in the development of seizures. Recently, Yue et al. demonstrated that MCC950 significantly reduced the levels of NLRP3 and the expression of ERS related markers in the hippocampi of pilocarpine-induced SE mice [73]. MCC950 has been shown to inhibit the NLRP3 inflammasome activation in KA-induced SE mice and KA-treated astrocytes [85]. Furthermore, clinical trials of MCC950 have also been initiated to evaluate its safety and efficacy [52]. MCC950 was initiated in a phase II clinical trial for rheumatoid arthritis, but it was discontinued due to liver toxicity.

CY-09 (4-[[4-Oxo-2-thioxo-3-[[3-(trifluoromethyl) phenyl] methyl]-5-thiazolidinylidene] methyl] benzoic acid) is a specific and direct inhibitor of NLRP3 that inhibited its ATPase activity and activation by binding to the Cys172 residue in the Walker A motif of the NACHT domain of NLRP3 [142]. CY-09 demonstrated favorable pharmacokinetic properties for safety, stability, and oral bioavailability. Previous studies have suggested that CY-09 could be used for the treatment of NLRP3 inflammasome-associated diseases, including type 2 diabetes, gout, thrombosis, cryopyrin-associated autoinflammatory syndrome (CAPS) mouse models, and other diseases [142,143]. Shen et al. reported that CY-09 inhibited the NLRP3 driven neuroinflammation in a PTZ-induced kindling mouse model, a chronic model of generalized seizures [144]. CY-09 repressed the expression of NLRP3, IL-1β, and IL-18 in injured brain tissue in the rat TBI models [145]. The findings from Wang et al. showed that CY-09 attenuates depression-like behaviors by inhibiting the NLRP3-mediated neuroinflammation in LPS-induced mice [146]. In a clinical trial, CY-09 was found to be effective in reducing the number and severity of seizures in people with focal epilepsy when compared to placebo. However, confirmatory studies are worth pursuing to broaden its potential in treating epilepsy.

Glyburide, a sulfonylurea also known as glibenclamide, is an FDA-approved ATP-sensitive K^+^ (K_ATP_) channel inhibitor to treat type 2 diabetes mellitus [147]. In 2001, Perregaux et al. reported for the first time that glyburide inhibits IL-1β release in LPS-activated human monocytes [148]. In another study, glyburide was shown to inhibit IL-1β release during bronchial hypo responsiveness through K_ATP_ channels [149]. Glyburide has been reported to exhibit anti-inflammatory effects mainly by the inhibition of microbial ligand-induced NLRP3 inflammasome activation and IL-1β secretion by blocking K_ATP_ channels [150]. Glyburide has been shown to block NLRP3 inflammasome activity and IL-1β secretion stimulated by islet amyloid polypeptide, which is associated with type 2 diabetes [151]. In human pancreatic islets, glyburide partially reduced the increased NLRP3 and IL-1β expression induced by LPS and ATP [152]. A recent study has reported that glyburide blocked the assembly and activation of NLRP3 inflammasome and IL-1β release by dampening the binding of NEK7 to NLRP3 in ventilator-induced lung injury [153]. Research has shown that it plays a dual role in attenuating cerebral edema and improving long-term cognitive function in a pilocarpine-induced mouse model of status epilepticus [154]. Acute administration of glyburide, 30 min prior to the PTZ, significantly increased the seizure threshold in an intravenous PTZ model of mice [155].

Beta-hydroxybutyrate (BHB), one of the ketone bodies, has been shown to reduce inflammatory cytokines’ release mediated by NLRP3. It has been studied for its potential therapeutic benefits in various inflammatory diseases, including epilepsy [156,157,158,159,160]. The mechanism of action by which BHB acts as an NLRP3 inhibitor is not fully understood. However, it is thought that BHB may inhibit NLRP3 inflammasome activation by modulating the production of reactive oxygen species (ROSs) and by reducing the levels of ATP in the cell. Beta-hydroxybutyrate also inhibits the NLRP3 inflammasome by preventing K^+^ efflux and reducing ASC oligomerization and speck formation [161]. Kim et al. reported that BHB reduced the spontaneous recurrent seizures in spontaneously epileptic Kcna1-null mice [162]. Furthermore, BHB decreased the seizure duration and frequency in a 6-Hz-induced seizure model of refractory epilepsy [163]. It is important to note that more research is needed to fully understand the mechanism of action and potential clinical applications of BHB as an NLRP3 inhibitor.

RRx-001 (1-bromoacetyl-3,3-dinitroazetidine) was initially developed as an anticancer agent by the aerospace industry [164], but has been extensively studied in in vitro and in vivo models of several inflammatory diseases including Alzheimer’s disease, stroke, multiple sclerosis, pulmonary fibrosis, and IBD [165]. In a randomized Phase 2 trial called PREVLAR; NCT03699956, RRx-001 administration in 53 first-line head-and-neck cancer patients dramatically improved the incidence, duration, time to onset, and severity of oral mucositis. RRx-001 has been safely evaluated in clinical trials, including in an ongoing phase 3 trial for the treatment of small cell lung cancer (REPLATINUM; NCT03699956) [164]. RRx-001 is a highly selective and the most clinically advanced small molecule NLRP3 inhibitor that has been safely evaluated in over 300 patients [164]. Mechanistically, RRx-001 covalently binds to cysteine 409 of NLRP3 on the central NACHT domain of NLRP3, which inhibits the assembly and activation of the NLRP3 inflammasome [164,165]. The BBB-penetrant nature of RRx-001 inhibitor and the preclinical assessment of this inhibitor in various neurodegenerative diseases [166] advances the possibility of this uncharged small molecule inhibitor being tested soon in epilepsy.

Several other classes of compounds that have been explored for NLRP3 inflammasome inhibition include flavonoids, chalcone, and boron-based compounds. Amentoflavone, a naturally occurring bioflavonoid [78], and semaglutide, a glucagon-like peptide-1 [74], were reported to affect epileptogenesis and reduce seizures via their neuroprotective effects because of NLRP3 inflammasome inhibition in PTZ-kindled mice. Sun et al. revealed that endogenous as well as exogenous IL-10 downregulates IL-1β production in microglia in mice exposed to epileptogenic injury thorough the STAT3-dependent inhibition of NLRP3 inflammasome activity [167].

Licochalcone B, Isoliquiritigenin, and Cardamomin are natural chalcone-based compounds that have shown promising NLRP3 inflammasome inhibitory effects. Licochalcone B binds with NEK7, preventing the interaction with NLRP3 which is important for NLRP3 inflammasome activation [168]. Isoliquiritigenin isolated from *Glycyrrhiza uralensis* has been reported to activate the Nrf2-mediated antioxidant signaling, preventing the activation of NF-κB and NLRP3 inflammasome [169,170]. Cardamomin reduced the protein levels of NLRP3, Casp1, and IL-1β in 2,4,6-Trinitrobenzenesulfonic acid (TNBS)-induced colitis mice [171]. BC7, BC23, and NBC6 are potent oxanorbornene molecules developed as potent NLRP3 inflammasome inhibitors. Among these three molecules, compound NBC6 showed the most potent inhibition of IL-1β release from THP-1 monocytes with an IC_50_ of 574 nM. BC7 and BC23 showed IC_50_ values of 1.16 μM and 2.29 μM, respectively, for the inhibition of IL-1β release [13,14].

Huperzine A, a naturally occurring sesquiterpene alkaloid and valproic acid, and one of the most prescribed medications against epilepsy, has been shown to inhibit activation of the NLRP3 inflammasome in the rat KA-induced model of epilepsy in a ROS-dependent manner [172]. Recently, furosemide (4-chloro-5-sulphonyl-N-furfuryl-anthranilic acid), a diuretic drug, has been shown to decrease the NLRP3 as well as NLRP1 level significantly when treated in combination with valproic acid in KA-induced epileptic rats [89].

Studies from Li and coworkers have reported that ibuprofen may have antiepileptic and neuroprotective effects in the rat model of PTZ-induced epilepsy via inhibiting NLRP3 inflammasome activation [173]. Rapamycin, an inhibitor of mTOR signaling, has been reported to alleviate the symptoms of seizures, anxiety, and depression in PTZ-kindled rats by inhibiting NLRP3 inflammasomes and ROS production [174]. Chaihu-Longgu-Muli decoction, a well-known ancient formula in traditional Chinese medicine, could significantly reduce the frequency and duration time of epileptic seizures, and inhibit the expression of NLRP3, TNF-α, Caspase-1, and IL-1β [175]. Natural products such as parthenolide and oridonin from *Rabdosia rubescens* were also able to inhibit the NLRP3 inflammasome [176,177]. It was observed that oridonin binds covalently with the Cys279 of the NLRP3 NACHT domain. This binding prevents the interaction between NLRP3 and NEK7, which is essential for NLRP3 inflammasome assembly and activation [176]. Recently, oridonin has been shown to rescue kanamycin-related hearing loss by inhibiting NLRP3 inflammasome activation [153,178]. BAY 11-7082, an NF-κB inhibitor, has also been reported to inhibit the NLRP3 ATPase activity in macrophages, independently of their inhibitory effect on NF-kB activity [177].

Curcumin ((1E,6E)-1,7-bis(4-hydroxy-3-methoxyphenyl)-1,6-heptadiene-3,5-dione)), a polyphenolic compound present in turmeric (*Curcuma longa*), exhibits antioxidant, anti-inflammatory, and neuroprotective properties, and its beneficial effect on epilepsy has been shown in many preclinical studies [179,180,181,182,183]. Curcumin has been reported to inhibit IL-1β release and prevent inflammation via the inhibition of NLRP3 [184] and suppressed KA-induced epileptic syndrome via inhibiting NLRP3 inflammasome activation in Sprague Dawley rats [68].

MicroRNAs (miRNAs) are endogenous ~20–23 nucleotide-long non-coding RNAs that bind to the 3′ untranslated region (3′UTR) of protein-coding mRNAs to regulate their translation and, thus, can have significant impacts on cellular processes. Several miRNAs that can target different components of the NLRP3 inflammasome and modulate its activity have been identified. For example, miR-223 has been shown to target the NLRP3 inflammasome in the brain and reduce neuroinflammation and neuronal damage [185], inhibit NLRP3 expression, and reduce inflammation in various disease models, including arthritis, atherosclerosis, and myocardial infarction [56,186,187]. miR-29c reduces the inflammatory response of microglia by modulating the NLRP3 inflammasome [188], miR-17-5p ameliorated NLRP3 inflammasome activation-mediated hypoxic–ischemic brain injury in rat [189], and miR-138-5p overexpression in epileptic neurons inhibits NLRP3 by directly binding with ubiquitin-specific peptidase 47 (USP47), a positive regulator of NLRP3 [190]. miR-29a-5p mimics protective TBI-induced BBB dysfunction via suppressing NLRP3 inflammasome activation [191]. Other miRNAs, such as miR-23a, miR-let-7e, miR-30e, and miR-223, have also been found to inhibit the NLRP3 inflammasome and reduce inflammation [192]. Overall, the use of miRNAs as NLRP3 inhibitors holds promise as a potential therapeutic approach for treating a range of inflammatory and autoimmune diseases. However, more research is needed to fully understand the mechanisms underlying miRNA regulation of NLRP3 and to develop effective and safe miRNA-based therapies for different diseases.

### 7.2. NLRP3 Inflammasome Inhibitors and Their Limitations as Remedial Strategies

NLRP3 inhibitors have been investigated as potential drugs for the treatment of various inflammatory and autoimmune diseases. Off-target effects are the significant drawback of using NLRP3 inhibitors. For example, MCC950 at high micromolar concentrations could inhibit carbonic anhydrase 2 [193] and block Cl^−1^ efflux from nigericin-activated macrophages [142]. CY-09 has been reported to affect cytochrome P450 enzymes [123]. Another inhibitor, oridonin, has several targets, e.g., AKT/2, c-Myc, p39, and MAPK [194]. Since NLRP3 is involved in many important cellular processes, including immune defense and tissue repair, blocking its activity could lead to unintended consequences including impairing pathogen clearance and thus increasing the risk of infection. MCC950 has been examined in a phase II clinical trial for the treatment of rheumatoid arthritis, but the trial was discontinued due to hepatic toxicity [52]. NLRP3 plays a crucial role in the immune system, and blocking its activity may also interfere with the body’s ability to fight infections and heal wounds.

Future studies should take advantage of available cryo-EM and crystal structures of NLRP3 bound to NEK7 [46], and focus on the development of structure-guided direct inhibitors with improved specificity and potency. Recently, Agarwal et al. rationally designed MCC950-derivative compounds [179,195,196]. These compounds were found to be potent and selective NLRP3 inhibitors with a good pharmacokinetic profile and high oral bioavailability in mice. In addition, another NLRP3 inhibitor, NT-0796, boasts innovative chemistry, delivering unparalleled potency and the promise of an extended pharmacodynamic impact. Furthermore, it demonstrates the capability to penetrate the BBB [197]. Since nanoparticle (NP)-based drug delivery is an emerging area of research in the field of nanomedicine and immunotherapy due to their intriguing properties such as target site specificity, systemic stability, and low toxicity [198], Mancuso et al. analyzed the effect of glyburide-loaded nanovesicles (GNVs) on NLRP3 inflammasome activation in a LPS and nigericin-activated THP-1 cell model [199]. Their results confirm that GNVs were able to inhibit IL-1β secretions more efficiently than free glyburide. Recently, Kulkarni and coworkers synthesized and analyzed MCC950-loaded nanoparticles (MCC NPs) and found that MCC NPs showed a significant reduction in IL-1β secretions in vitro and in vivo [200]. Tang et al. engineered a unique delivery system, VHPK-PLGA@COL, incorporating colchicine and demonstrating enhanced biosafety and prolonged drug release. This was validated both in cell culture and in animal models [201]. The inhibitory effect of VHPK-PLGA@COL on NLRP3 and its downstream molecules was more significant than that of free colchicine. Exosome-like nanoparticles from ginger rhizomes strongly inhibited NLRP3 inflammasome activation [202]. Another group prepared garlic chive-derived vesicle-like nanoparticles which exhibit potent anti-NLRP3 inflammasome activity [203]. Moreover, dexamethasone-loaded ROS-responsive polymer nanoparticles prepared by a modified emulsion approach had a strong ability to inhibit the expression of NLRP3, caspase1, and IL-1β [204]. It has been reported that nanoparticles themself trigger NLRP3 inflammasome activation [205,206,207,208]. However, the formation of a protein corona layer on lipid NPs caused a significant reduction in NLRP3 inflammasome activation and controlled the toxicity, biodistribution, and cellular uptake [209]. Chalcones are natural compounds with an α, β unsaturated carbonyl group (Michael acceptor) found in many plants and have gained attention for their medicinal properties [210,211]. We and others investigated some chalcones for their potential as NLRP3 inhibitors [212,213]. Our preliminary data encourage the further development of more potent NLRP3 inhibitors based on this chalcone scaffold, which could lead to the development of novel treatments for epilepsy and other inflammatory diseases.

**Table 1 ijms-25-06078-t001:** NLRP3 inhibitors as potential epilepsy therapeutics at pre-clinical stages.

Compound	Nature of Inhibitor	Mechanism of Action	Disease Model	References
MCC950	Sulfonylurea	Directly interacts with Walker B motif of the NACHT domain, changes NLRP3 conformation. Blocks NLRP3-dependent ASC oligomerization and NLRP3 inflammasome activation Blocks ATPase activity	In vitro SH-SY5Y model and in vivo model of cerebral trauma induced by PTZ Pilocarpine-induced SE mice KA-induced SE mice	[72,73,85,126,127,131,137]
CY-09	Glitazones	Directly binds to Cys172 residue in the Walker A motif of NLRP3 NACHT domain and inhibits NLRP3 ATPase activity	0.83 mg/kg LPS-induced mice PTZ induced kindling mouse model	[144,146,214]
Glyburide	Sulfonylurea	Suppresses K_ATP_ channels and inhibition of ASC agglomeration Blocks the assembly and activation of NLRP3 inflammasome and IL-1β release by dampening the binding of NEK7 to NLRP3	Pilocarpine-induced mouse model of SE Seizures induced by i.v. or i.p. PTZ models	[149,154,155]
BHB	Natural products	Inhibition of K^+^ efflux and reduced ASC oligomerization and speck formation Inhibit NLRP3 inflammasome activation by modulating the production of ROS and by reducing the levels of ATP in the cell.	Epileptic Kcna1-null mice 6-Hz induced seizure model of refractory epilepsy	[161,162,163]
Amentoflavone	Naturally occurring bioflavonoid	Exerts neuroprotective effects by inhibiting the NLRP3 inflammasome	The chronic epilepsy model and BV2 microglial cellular inflammation model were established by PTZ kindling or LPS stimulation, respectively.	[78]
Semaglutide	Glucagon like peptide-1	Decreases seizure severity, alleviated hippocampal neuronal apoptosis, ameliorated cognitive dysfunction by blocked ASC oligomerization and NLRP3 inflammasome activation	PTZ-kindled C57/BL6J mouse model and LPS induced inflammation in BV2 cells	[74]
Huperzine A	Naturally occurring sesquiterpene	Inhibits activation of NLRP3 inflammasome in a ROS-dependent manner	Rat KA-induced model of epilepsy	[172]
Furosemide	Sulfonamide	Increases the efficacy of valproic acid by inhibiting NLRP3 inflammasome activation	KA-induced epileptic rats	[89]
Ibuprofen	nonsteroidal anti-inflammatory drug (NSAID)	Exhibits antiepileptic and neuroprotective effects via inhibiting NLRP3 inflammasome activation	Rat model of PTZ-induced epilepsy	[173]
Rapamycin	Macrolide compound	Inhibits NLRP3inflammasome and ROS production	PTZ-kindled rats	[174]
Chaihu-Longgu-Muli decoction	Traditional Chinese medicine	Could significantly suppress the frequency and duration time of epileptic seizures via reducinge the expression of NLRP3, Caspase-1 TNF-α and IL-1β.	Rats with TLE	[175]
Parthenolide	Naturally occurring sesquiterpene lactone	Supresses NLRP3 ATPase activity by alkylating cysteine residues in ATPase domain of NLRP3 Inhibits protease activity of caspase 1	In vitro LPS and ATP induced NLRP3 stimulation	[177]
Bay 11-7082	Sulfone	Blocks ATPase activity of NLRP3 (Juliana et al., 2010)	In vitro LPS and ATP induced NLRP3 stimulation	[177]
Oridonin	Natural terpenoids	Binds to Cys279 of NLRP3 NACHT domain and inhibits the interaction between NLRP3 and NEK7 thereby inhibiting the NLRP3 inflammasome activation	TBI mice	[176,215]
Curcumin	Natural polyphenolic compound	Inhibit IL-1β release and prevent inflammation via inhibition of NLRP3	KA-induced epileptic syndrome in Sprague Dawley rats	[68,184]

MCC950, 1-(1,2,3,5,6,7-Hexahydro-s-indacen-4-yl)-3-[4-(2-hydroxypropan-2-yl)furan-2-yl]sulfonylurea; NACHT, NAIP (neuronal apoptosis inhibitor protein), C2TA (MHC class 2 transcription activator), HET-E (incompatibility locus protein from Podospora anserina) and TP1 (telomerase-associated protein); NLRP3, NLR family pyrin domain containing 3; ASC, apoptosis-associated speck-like protein containing a CARD; PTZ, pentylenetetrazol; kA, kainic acid; TLE, Temporal lobe epilepsy; SE, status epilepticus, LPS, lipopolysaccharide; CY-09, 4-[[4-Oxo-2-thioxo-3-[[3-(trifluoromethyl) phenyl] methyl]-5-thiazolidinylidene] methyl] benzoic acid; NEK7, NIMA related kinase 7; BHB, beta-hydroxybutyrate; NSAID, nonsteroidal anti-inflammatory drug; ROS, reactive oxygen species; TBI, traumatic brain injury; ATP, adenosine triphosphate; IL-1β; interleukin-1beta.

## 8. Conclusions and Perspectives

Scientific evidence supports the role of NLRP3 in epileptic seizures and the use of NLRP3 inhibitors seems promising in understanding the biological mechanisms behind epilepsy, leading to the development of biomarkers for early detection and more targeted, effective treatments to reduce seizures in high-risk individuals with epilepsy. Clinical trials are underway to investigate the efficacy and safety of these inhibitors in humans, and if these trials are successful, NLRP3 inhibitors may eventually become standard medical care for patients with epilepsy.

However, there is still much to be investigated before NLRP3 inhibitors are officially approved for the treatment of epilepsy.

## Figures and Tables

**Figure 1 ijms-25-06078-f001:**
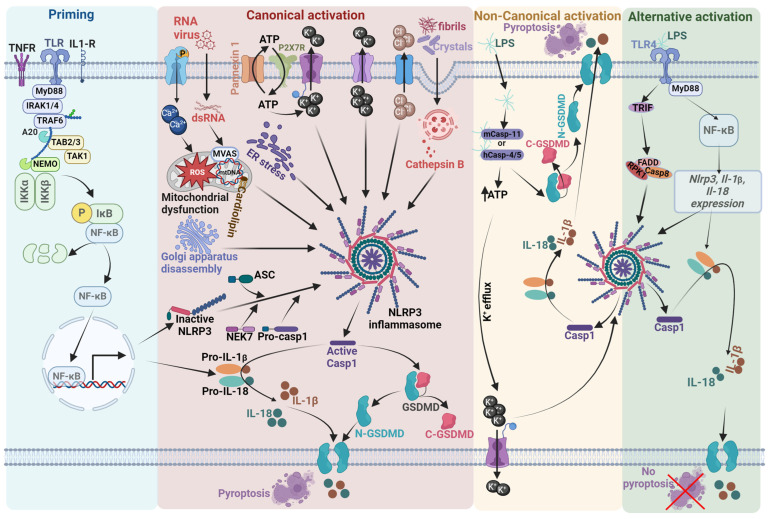
Schematic illustration of the mechanism regulating NLRP3 inflammasome activation in canonical, non-canonical, and alternate pathway. Optimal activation of NLRP3 requires two steps. The first step is priming, which is initiated by extracellular PAMPs and endogenous cytokines by the PRRs, which then upregulates the NF-κB-mediated transcription of NLRP3, pro-IL-1β, and pro-IL-18. The second step is activation, which includes canonical and non-canonical pathways. The canonical pathway is triggered by multiple pathogens and inflammatory agents through a combination of important and related events such as activation of the purinergic receptor P2X7 by ATP, cathepsin release following lysozyme rupture, opening of Ca2+ channels to allow ion flux, mitochondrial dysfunction, ROS formation, Golgi apparatus disassembly, and endoplasmic reticulum stress. Once activated, oligomerization of the NLRP3 inflammasome is thought to induce conformational changes that generate active caspase-1, which converts pro-pro-IL-1β and pro-IL-18 to mature bioactive IL-1β and IL-18. Additionally, Casp1 cleaves the protein gasdermin D to generate N-terminal gasdermin D to form pores, allowing IL-1β and IL-18 to leave the cell and effectively execute a highly inflammatory form of cell death that is termed pyroptosis. In the non-canonical pathway, intracellular LPS is directly recognized by the CARD domain of caspase-11 in mice and caspase-4/5 in humans, ultimately leading to IL-1β and IL-18 release through the activation of the NLRP3-ASC-Casp1 pathway. The alternative pathway of activation is caused by TLR4 agonists like LPS, which activates the TLR4-TRIF-RIPK1-FADD-Casp8 signaling pathway. Casp8 activates the NLRP3 inflammasome but lacks ASC speck formation, pyroptosis induction, or K+ efflux.

**Figure 2 ijms-25-06078-f002:**
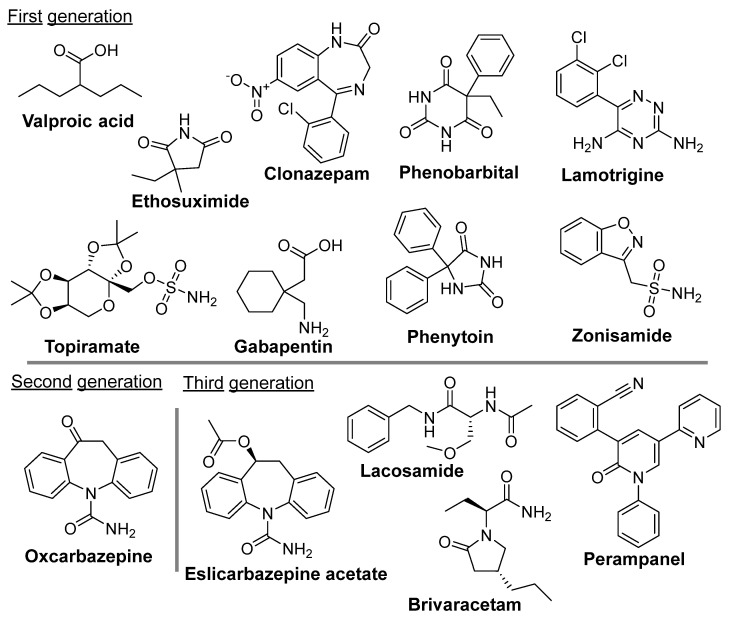
Chemical structures of clinically approved antiseizure medications discussed in this review.

**Figure 3 ijms-25-06078-f003:**
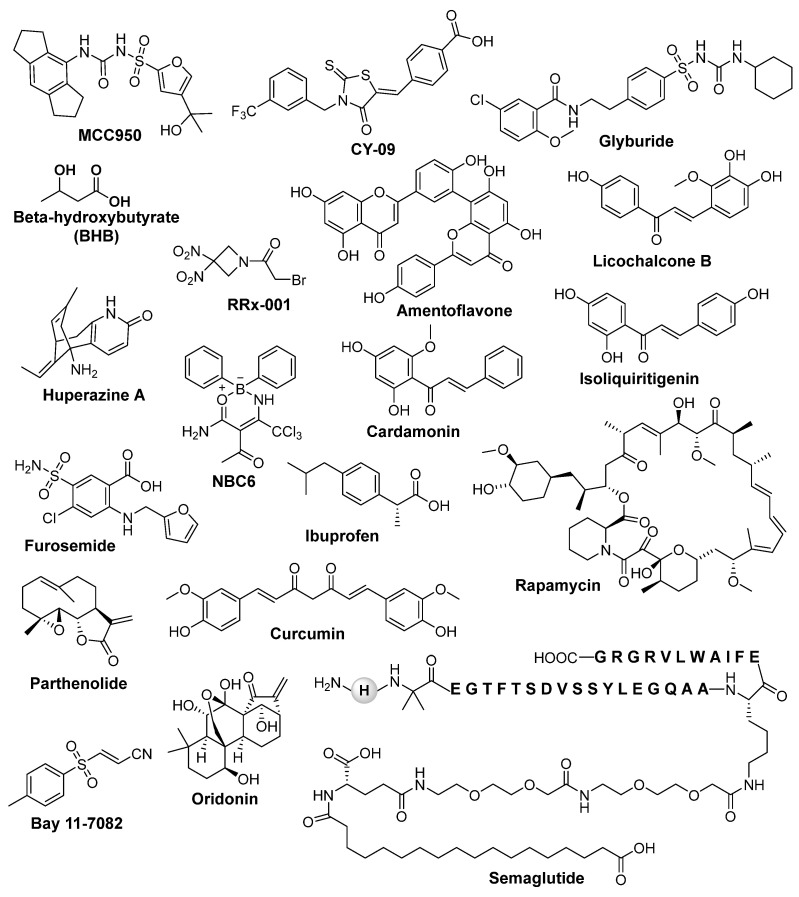
Chemical structure of different NLRP3 inflammasome inhibitors discussed in this review.

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
