# Peer review of "NLRP3 Inflammasome Inhibitors for Antiepileptogenic Drug Discovery and Development"

_ijms, 2024, doi:10.3390/ijms25116078_

Round 1

Reviewer 1 Report

Comments and Suggestions for Authors

The manuscript “NLRP3 inflammasome inhibitors for antiepileptogenic drug discovery and development” by Drs. Inamul Haque et al the authors observed and reviewed the results of modern studies on the nature of epileptogenesis and its relationship with the NOD-like receptor family

containing NLRP3, mechanisms of their activation and their inhibitors, as well as their prospects for treatment epilepsy.

 The authors have done a great job, the review presented is extensive and informative. The information is logically connected, I have no fundamental objections. However, there are some questions that I would like to clarify.

As the authors rightly note:

LINE 188: ...the NLRP3 inflammasome components are highly expressed in neurons, astrocytes, and microglia in the epileptogenic zone of patients with temporal lobe epilepsy (TLE), a form of epilepsy associated with hippocampal neuronal atrophy and in the hippocampi of a mouse model of status epilepticus (SE)-induced by pilocarpine...

How specific is this process? Is there a connection between this expression and other proteins?

Another protein called activity-regulated cytoskeleton-associated protein (arc) is also intensely expressed in the zone of epileptic activity (Sibarov et al, 2023; DOI: 10.3389/fneur.2023.1201104). Is there any data on the interaction of these peptides (and their genes)? The authors' opinion would be very valuable for the audience.

Channel protein Pannexin-1 (Panx1) expression is raised in epileptic neurons, it is likely related with elevated levels of extracellular K+. However, Panx1, like some other channel proteins, can be regulated through NLRP3 (doi: 10.3390/biom13030505). Could this be the basis for any new therapeutic strategies in the treatment of epilepsy?

Minor criticism: I would arrange the keywords in alphabetical order

The presentation of a subject is systematic and comprehensive, list of references is quite full. I am happy to recommend the manuscript for the publication after minor corrections mentioned above.

Reviewer 2 Report

Comments and Suggestions for Authors

This is a well-written review on the development of NLRP3 inhibitors for the treatment of epilepsy. The topic of this review is interesting and exhaustively discussed.  The figure and the table are properly constructed and clearly described. All presented facts are supported by adequately chosen references. I have only one minor remark. This manuscript emphasizes the potential of NLRP3 inhibitors in suppressing epileptic seizures, not epileptogenesis as stated in the title. It is unclear if NLRP3 inhibitors are the best candidates for antiepileptogenic drugs or for new class of ASMs.
